# Analysis of the Differences in Volatile Organic Compounds in Different Muscles of Pork by GC-IMS

**DOI:** 10.3390/molecules28041726

**Published:** 2023-02-11

**Authors:** Shengnan Duan, Xiaoyan Tang, Wusun Li, Xinyuan Huang

**Affiliations:** 1Key Laboratory of Meat Processing and Quality Control, College of Food Science and Technology, Nanjing Agricultural University, Nanjing 210095, China; 2Key Laboratory of Agro-Product Quality and Safety, Institute of Quality Standards & Testing Technology for Agro-Products, Chinese Academy of Agricultural Sciences, Beijing 100081, China

**Keywords:** gas chromatography-ion mobility spectrometry (GC-IMS), volatile organic compounds (VOCs), principal component analysis (PCA), partial least squares-discriminant analysis (PLS-DA)

## Abstract

As the main consumed meat of Chinese residents, pork has a unique flavor, but the internal volatile organic compounds that cause the flavor differences between pork muscles are not clear at present. In this study, four muscles of Duroc × (Landrace × Yorkshire) pigs (loin, ham, shoulder and belly) were used as experimental subjects. Through the analysis of volatile organic compounds in four muscles of pork, the internal volatile organic compounds of different muscles of pork were discussed. Gas chromatography-ion mobility spectrometry was employed to analyze the four muscles, and volatile organic compounds in these muscles were analyzed and identified. A total of 65 volatile organic compound peaks were obtained by gas chromatography-ion mobility spectrometry. From the qualitative database, a total of 49 volatile organic compounds were identified, including aldehydes, alcohols and ketones. With the variable importance for the projection greater than 1 and significance level less than 0.05 as the criterion, the organic compounds with significant differences were screened by partial least squares-discriminant analysis and significance difference analysis. It was determined that 2-pentylfuran, 2-butanone (M), pentanal (M), butanal (D), (*E*)-2-hexenal, (*E*)-2-heptenal (D), 1,2-propanediol and 2-methylpropanal were the differential organic compounds that distinguish the four pork muscles.

## 1. Introduction

As the main meat consumed by Chinese residents, pork produced in China in 2021 exceeded 50 million tons, among which Duroc × (Landrace × Yorkshire) pigs (DLYs) accounted for more than 90% of the pork market because of their fast growth and high lean meat rate. Pork is rich in high-quality protein, fatty acids, essential amino acids and trace elements, possessing high nutritional value [1]. With economic development and the improvement of people’s living standards, consumers are paying more attention to the texture and flavor of pork [2]. In the consumer market, pork loin, ham, shoulder and belly are four common muscles of pork, which can provide consumers with different flavors, but the internal flavor substances that produce their different flavors are still unclear.

Flavor is an important edible quality of meat. The volatile flavor of meat is mainly caused by internal volatile organic compounds (VOCs). VOCs are mainly generated through lipid oxidative degradation and Maillard reaction [3]. Oxidative degradation of lipids will produce aldehydes, ketones, hydrocarbons and other volatile flavor compounds [4]. The Maillard reaction will produce furan, furanone and other heterocyclic compounds [5]. At present, pork research mainly focuses on the processing technology [6,7] and physical and chemical quality [8,9,10] of pork, and research on pork flavor is relatively scarce. In terms of pork flavor, most researchers pay more attention to the flavor of pork products [11,12,13] and rarely study the flavor of different muscles of pork. Therefore, by studying different muscles of pork, we can clarify their flavor differences and explore the related influencing factors.

Gas chromatography-ion mobility spectrometry (GC-IMS) effectively combines the efficient separation ability of gas chromatography with the advantages of a fast response and low cost of ion mobility spectrometry, so as to separate and identify samples according to the differences in the mobility of electrons in the electric field of different samples [14,15]. Most organic compounds have high electronegativity or high proton affinity [16], which can be effectively identified by GC-IMS [2]. GC-IMS can visually display volatile organic differences among samples by forming 2D fingerprint spectra [17]. Moreover, compared with gas chromatography-mass spectrometry (GC-MS), GC-IMS does not require high vacuum conditions and is characterized by faster detection and more intuitional results. Therefore, in recent years, GC-IMS has been widely used in food detection and rapidly applied to the evaluation of food flavor and determination of VOCs. For example, Li et al. [18] used GC-IMS to measure flavor substances in boiled salted duck, and identified 50 volatile organic compounds, including aldehydes, alcohols, esters, ketones, hydrocarbons, etc. These compounds were analyzed by GC-IMS and characterized by retention index and drift time. The retention index was referred to the NIST library, and the drift time was determined by the standard of each compound. However, of the 50 volatile substances, 15 of them were not identified as the database was limited.

In this study, headspace-gas chromatography-ion mobility spectrometry (HS-GC-IMS) was used to analyze and identify the VOCs in pork loin, pork ham, pork shoulder and pork belly of Duroc × (Landrace × Yorkshire) pigs. Using multivariate statistical methods such as principal component analysis (PCA) and partial least squares discriminant analysis (PLS-DA), the VOCs differences of the four muscles were analyzed to determine the characteristic organic substances in the four muscles of pork.

## 2. Results and Discussion

### 2.1. GC-IMS 2D Spectrum of Volatile Organic Compounds from Different Pork Muscles

HS-GC-IMS was used to determine VOCs in pork loin, pork ham, pork shoulder and pork belly of DLYs, with the Reporter plug-in employed to obtain the two-dimensional (2D) spectrum. As shown in Figure 1, the entire spectrum represented all VOCs in the four muscles. The red line on the left represents the reactive ion peak (RIP), the abscissa represents the migration time, and the ordinate represents the retention time (RT). As shown in Figure 1, the drift time was 1.0–2.0 ms, and the retention time was mainly between 100 and 900 s. With each dot standing for a VOC, the concentrations of the VOCs can be represented by the depth of color in the 2D spectrum [19]. The redder the dot, the higher the concentration; the whiter the dot, the lower the concentration. As can be seen from Figure 1, there were obvious differences in the VOCs in the four muscles of pork. There were more volatile organic substances highlights in pork loin and ham, while there were fewer volatile organic substances highlights in pork shoulder and belly, indicating that there were more VOCs in loin and ham than in the shoulder and belly.

In order to perceive the difference in the VOCs in the four muscles more obviously, the 2D spectrum of pork loin was selected by the difference comparison option of the Reporter plug-in as the reference, and the reference was deducted from the spectrum of the other three muscles to obtain the 2D difference comparison spectrum of GC-IMS. If the VOCs in the four muscles were consistent, the background after deduction would be white, the red color meant that the concentration of the substances was higher than the reference, and the blue color meant that the concentration of the substances was lower than the reference [20]. As shown in Figure 2, the overall volatile organic content of pork ham was higher than that of pork loin, while the overall volatile organic content of pork shoulder and belly was lower than that of pork loin.

### 2.2. Fingerprints of Volatile Organic Compounds in Different Muscles of Pork

In order to construct the fingerprint of the VOCs in four muscles of pork, including loin, ham, shoulder and belly, and analyze the differences of the VOCs in the four muscles, the NIST and IMS databases of GC-IMS were used for qualitative analysis of the VOCs in the four muscles of pork. It should be noted that in the qualitative process, due to the high proton affinity of the compounds [21,22,23], the compounds with a higher concentration may generate multiple signals and spots, which are named monomer (M), dimer (D) or polymer (P) [24,25]. The Gallery Plot plug-in was used to generate the fingerprint from the signal values of all volatile compounds in the GC-IMS spectrum measured for the four muscles. As shown in Figure 3, the abscissa was represented for the qualitative name of VOCs, and the ordinate was the sample information of pork. Each row in Figure 3 represented all the signal peaks selected from a sample, and each column represented the signal peaks of the same volatile organic compounds in different samples. The sequence of fingerprints from left to right was a manual arrangement of the same pattern of substances, not a peak chronological order. Figure 3 showed the complete volatile organic com-pounds information for each pork muscles and the differences in volatile organic compounds between pork muscles. 

As shown in Figure 3, the VOCs of pork loin and pork ham were significantly different from those of pork shoulder and pork belly in region A. The compounds involved were mainly 23 VOCs, such as nonanal (M), nonanal (D), benzaldehyde (M), benzaldehyde (D), 3-methyl butanal (D), (*E*, *E*)-2,4-heptadienal and 1-octen-3-ol. The contents of these 23 compounds were relatively high in pork loin and pork ham, but relatively low in pork shoulder and pork belly.

The compounds involved in region B were 2-octanol (M), (*E*)-2-heptenal (D) and acetoin. The content of these compounds was relatively low in pork loin, but relatively high in pork ham. The VOCs in regions C, D and E were mainly allylacetic acid, propionic acid, 1,2-propanediol, isopropyl alcohol, 2-ethylfuran, 2-methylpropanal, butanal (D), tert-butyl methyl ether, ethyl acetate and another 16 compounds, but only existed in pork ham, pork shoulder and pork belly. It is speculated that it was caused by individual pig differences. According to the sample information displayed in the vertical coordinate, the samples with abnormal data were ham 1, shoulder 2 and belly 2, which were from the first and second pigs. Therefore, in the subsequent analysis, the data of pigs 1 and 2 were excluded, and only the data of the remaining four pigs were analyzed.

### 2.3. Qualitative Analysis of Volatile Organic Compounds in Different Pork Muscles by GC-IMS

The qualitative analysis of GC-IMS is based on the retention index (RI) of gas chromatography and the relative migration time of IMS. The IMS database was established by Shandong Haineng Scientific Instrument Co., Ltd. (Shangdong, China) through standard products. In this study, there were more than 520 kinds of substances, but some substances still could not be characterized, as shown in Figure 3, they were expressed by serial numbers. In addition, 65 peaks of VOCs were detected by GC-IMS in pork and 49 VOCs were identified according to the database. The information of flavor compounds qualitatively obtained from the database was shown in Table 1. There were 18 aldehydes, 10 alcohols, 6 ketones, 5 esters, 2 ethers, 3 acids, 2 heterocyclics, 3 hydrocarbons, including 39 monomers (M) and 10 dimers (D).

As can be seen from Figure 4, aldehydes occupy the largest proportion (about 40%) in pork organic compounds, and aldehydes have a relatively low threshold and are volatile, which has a great influence on aroma [26]. In particular, unsaturated aldehydes participate in the interaction between amino acids and carbonyls as an important intermediate in lipid oxidative degradation, and are the main VOCs that form in the process of lipid oxidation of pork [27]. Aldehydes can also be produced through protein oxidation [27]. For example, benzaldehyde is produced by the Strecker degradation reaction of phenylalanine degradation products. However, benzaldehyde has an unpleasant odor and affects the aroma of pork [27]. As shown in Table 1, the peak volumes were represented for the contents of organic compounds. The peak volumes of pentanal(M), pentanal(D), heptanal, octanal, nonanal(M), 3-methyl-butanal(M), 3-methyl-butanal(D) and hexanal were larger in those of aldehydes, which indicated that the concentrations of these compounds were higher. As can be seen from Table 1, the contents of most aldehydes in pork loin and ham were higher than those in pork shoulder and belly. For example, the content of octanal and nonanal in pork loin and ham was higher than that in other muscles. These two aldehydes are oxidation products of linoleic acid [28], showing an oil flavor. As can be seen from Figure 5, among the four muscles of pork, pentanal (D) was an aldehyde of the highest content, and the content of pentanal (D) in pork ham was significantly higher than that in other muscles (*p* < 0.05). The pentanal in pork can produce a unique bread flavor and fruit flavor. The content of hexanal in the four muscles was relatively high, and the content of hexanal in pork ham was significantly higher than that in the other muscles (*p* < 0.05). It is a common VOC in meat products, with a charming grass flavor [29].

In the organic compounds of pork, esters accounted for more than 20%, and the content of esters in the four muscles of pork ranked second. Esters are produced by esterification reactions between acids resulting from the degradation of fats or proteins and alcohols [27]. As can be seen from Table 1, the content of ester compounds in pork ham was significantly higher than that in the other three muscles (*p* < 0.05), such as ethyl acetate and ethyl butyrate. Although the content was high in the four muscles, due to their high threshold, they have little contribution to the formation of pork flavor [30]. The content of alcohols in the four muscles of pork was slightly low, about 12%, but alcohols play a coordinating role in the formation of the overall flavor of meat products [27]. Alcohols are mainly produced by the degradation of linoleic acid in muscle by lipoxygenase and peroxidase [27]. Most of them exhibit sweet, fresh, fruit and vegetable fragrance, flower fragrance or other pleasant smells, which can increase the volatile flavor of meat products [27]. Alcohols can be divided into saturated alcohols and unsaturated alcohols. Saturated alcohols have a higher threshold and make relatively less contribution to the flavor, while unsaturated alcohols have a lower threshold and exert an important impact on the overall flavor [28]. For example, 1-octen-3-ol is an unsaturated alcohol with a low threshold and strong mushroom flavor, which may be the main provider of pork aroma [26]. Ketones contained in the four muscles were about 7%, which are usually related to the flavor characteristics of cream and fruit, such as 2-heptanone produced by the degradation of linoleic acid, which has a sweet smell of cheese [31]. As shown in Table 1, the content of 2-heptanone in pork ham was significantly higher than that in the other muscles. Heterocyclic compounds occupied a relatively low proportion of the flavor compounds in the four muscles of pork, with only two heterocyclic compounds, but heterocyclic compounds are an important compound of pork flavor because they have a low threshold and have a strong odor at low concentrations. For example, the content of 2-pentylfuran produced by the oxidation of linoleic acid was significantly higher in pork loin than in the other muscles (*p* < 0.05), which has the scent of fruit and butter and contributes significantly to the formation of pork aroma [32,33].

### 2.4. Principal Component Analysis of Volatile Organic Compounds in Pork

PCA is an unsupervised data analysis method in which sample variance information can be obtained through PCA. Origin software was used for PCA of volatile flavor substances in four muscles of pork. The PCA scores plot is shown in Figure 6, and the loadings plot of the compounds is shown in Appendix A, and the corresponding numbers of VOCs in the loadings plot are shown in Appendix A. As shown in Figure 6, the contribution rates of PC1 and PC2 were 45.5% and 19.7%, respectively, and the cumulative variance contribution rate of the two principal components was 65.2%, which could reflect the volatile organic difference of the four muscles of pork. As can be seen from Figure 6, the sample points from the same muscle of pork were roughly clustered together, among which pork loin and pork ham were significantly separated from pork shoulder and pork belly, indicating that VOCs of the former two muscles and the latter two muscles were varied distinctly. The occurrence in pork loin and pork ham partially overlapped, but the sample points were clearly separated, indicating that VOCs between these two pork muscles was relatively similar, but there were some differences on the whole. The ranges in pork shoulder and pork belly also overlapped, indicating that VOCs similarity between pork shoulder and pork belly, but there was still a difference between the two pork muscles.

### 2.5. Partial Least Squares Discriminant Analysis of Volatile Organic Compounds in Pork

PLS-DA is a supervised data analysis method, through which the variable importance for the projection (VIP) data analysis can be obtained and differentiated substances can be identified. SIMCA data analysis software was used to conduct PLS-DA on the VOCs in four muscles of pork, and the organic compounds with significant differences were screened according to the variable importance for the projection (VIP) diagram. It is generally believed that the compounds with VIP > 1 are responsible for the differences in the samples [34,35]. According to the PLS-DA results, VIP values of all VOCs are shown in Appendix A, among which 20 compounds with VIP > 1, and 16 compounds were qualitatively determined against the GC-IMS database. As shown in Figure 7, the 16 organic compounds were sorted from the largest to the smallest according to the VIP value, including 1,2-propanediol, 2-octanol (M), 2,3-butanediol, acetoin, (*E*)-2-heptenal (D), 2-pentylfuran, (E)-2-hexenal, pentanol (M), (*E*)-2-heptenal (M), 2,3-butanedione, allylacetic acid, 2-ethylfuran, butanol (D), 2-methylpropanal, 1-pentanol (D) and 2-butanone (M).

With both VIP > 1 in Figure 7 and the significant difference in Figure 5 (*p* < 0.05) considered, the 16 different organic compounds qualitatively obtained were analyzed. It was found that the concentrations of 2-pentylfuran and 2-butanone (M) in pork loin were significantly higher than those in other muscles (*p* < 0.05), and the concentrations of pentanal (M) and butanal (D) in pork ham were significantly higher than those in other muscles (*p* < 0.05). The concentrations of (*E*)-2-hexenal, (*E*)-2-heptenal (D), 1,2-propanediol and 2-methylpropanal were significantly higher in pork belly than in other muscles (*p* < 0.05). Therefore, it is concluded that the eight flavor compounds of 2-pentylfuran, 2-butanone (M), pentanal (M), butanal (D), (*E*)-2-hexenal, (*E*)-2-heptenal (D), 1,2-propanediol and 2-methylpropanal are the different organic compounds that distinguish the four pork muscles.

## 3. Materials and Methods

### 3.1. Materials and Chemicals

Pork samples from different muscles of six 180-day-old DLY pigs of 80 kg carcass weight under the same feeding conditions were collected from Beijing Shunxin Peng-cheng Co., Ltd. (Beijing, China). After slaughter and cooling at 4 °C for 24 h, four muscles from the left half of the carcass, namely the loin, ham, shoulder and belly, were collected, the fascia was removed, and then they were placed in a −20 °C refrigerator for freezing.

C_4_-C_9_ normal ketones were purchased from Shandong Haineng Scientific Instrument Co., Ltd. (Shangdong, China).

### 3.2. Sample Preparation

The meat was homogenized and mixed in a meat grinder, stored at −20 °C, and thawed at 4 °C for 12 h before analysis. Then, 3.00 g meat samples were weighed in a 20 mL headspace bottle and heated in a constant temperature water bath at 100 °C for 15 min prior to GC-IMS analysis, with three parallels for each sample.

### 3.3. GC-IMS Analysis

The VOCs in different muscles of DLYs were identified using a gas chromatography-ion mobility spectrometry (FlavourSpec^®^, Shandong Haineng Scientific Instrument Co., Ltd., China) equipped with a SE-5 weakly polar capillary column (15 m × 0.53 mm × 1 μm; Restek, Bellefonte, PA, USA).

Before analyzing the samples, the instrument was calibrated with C_4_-C_9_ normal ketones. The prepared samples were incubated at 60 °C for 20 min. Then, 500 μL of headspace was automatically injected into a heating syringe at 85 °C. The column temperature was set as 60 °C and the drift tube temperature was set as 45 °C. Nitrogen gas (purity ≥ 99.999%) was used as the carrier gas, with the flow rates starting from 2 mL/min, then increased to 10 mL/min for 0~10 min, and finally increased to 100 mL/min for 10~25 min. Nitrogen gas (purity ≥ 99.999%) was also used as the drift gas with a constant flow rate of 150 mL/min.

### 3.4. Statistical Analysis

The volatile organic data of different pork muscles were collected and analyzed by instrumental analysis software VOCal, GC-IMS Library Search and built-in plug-ins (Reporter and Gallery Plot). GC-IMS data were qualitatively analyzed using NIST and IMS databases built into the Library Search software. The Reporter and Gallery Plot plug-ins of GC-IMS were employed to compare volatile organic fingerprints of different pork muscles. Statistical analysis was performed using SPSS 26.0 software (IBM, Chicago, IL, USA), Additionally, the data obtained were subjected to one-way analysis of variance (ANOVA), and Duncan’s multiple comparison was used to determine the difference between the four muscles of a compound, with the significance level defined as *p* < 0.05. PCA and PLS-DA were used to analyze the peak volume of volatile organic compounds in four pork muscles. The peak volume data of volatile organic compounds determined by GC-IMS was imported into Origin 2022 (OriginLab, Northampton, MA, USA), and the plug-in of principal component analysis was used in the software to standardize the peak volume data. The differences of the four muscles were analyzed by PCA. The peak volume of volatile organic compounds of the four pork muscles determined by GC-IMS was imported into SIMCA14.1 (Umetrics, Malmo, Sweden). After data standardization, PLS-DA was used to determine the VOCs with VIP > 1. 

## 4. Conclusions

In order to investigate the differences in the volatile organic compounds (VOCs) in four muscles of pork, headspace gas chromatography-ion mobility spectrometry (HS-GC-IMS) was used to analyze and identify the VOCs in them. A total of 65 VOCs peaks were obtained by GC-IMS. A total of 49 VOCs were identified, including 18 aldehydes, 10 alcohols, 6 ketones, 5 esters, 2 ethers, 3 acids, 2 heterocyclic substances and 3 hydrocarbons.

According to PCA, there were some differences in the four muscles of pork. Through PLS-DA analysis and significance difference analysis, VIP > 1 and *p* < 0.05 were used as the criteria to select significantly differential organic compounds. After data analysis, the significantly differential organic compounds in the four muscles of pork were identified as 2-pentylfuran, 2-butanone (M), pentanal (M), butanal (D), (*E*)-2-hexenal, (*E*)-2-heptenal (D), 1,2-propanediol and 2-methylpropanal. It was concluded that these compounds were the differential VOCs that distinguish the four pork muscles.

## Figures and Tables

**Figure 1 molecules-28-01726-f001:**
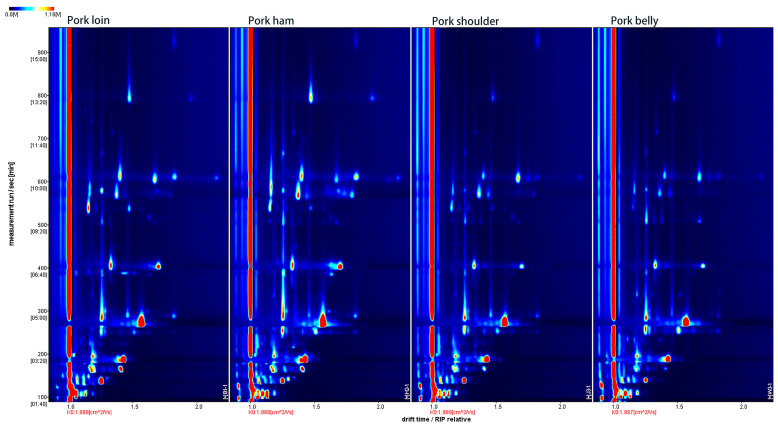
Two-dimensional GC-IMS spectra of volatile organic compounds in four muscles of pork.

**Figure 2 molecules-28-01726-f002:**
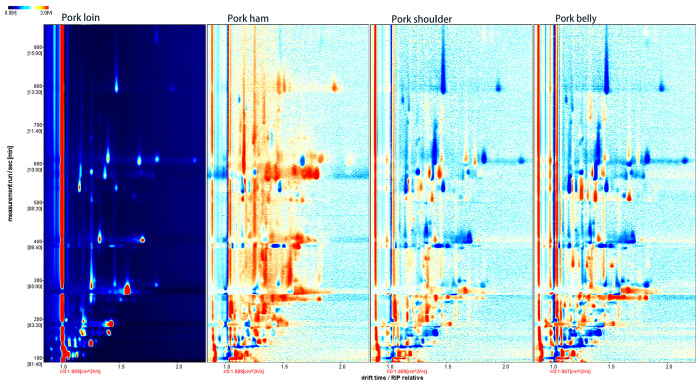
Comparison of differences in 2D GC-IMS spectra of volatile organic compounds in four muscles of pork.

**Figure 3 molecules-28-01726-f003:**
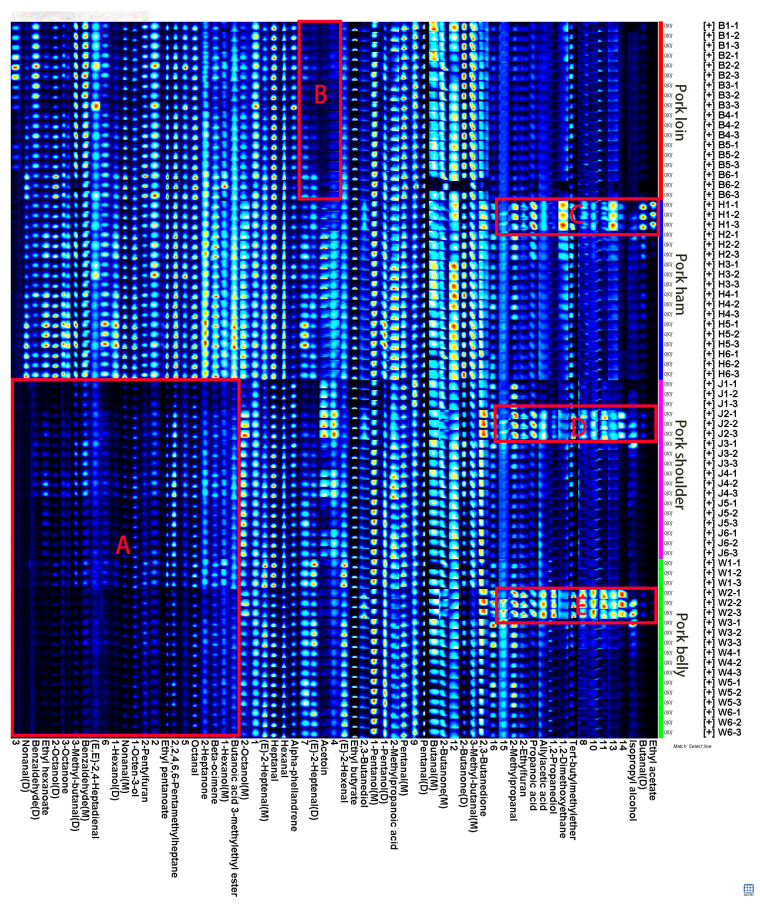
Fingerprints of volatile organic compounds in four muscles of pork (note: B represents for pork loin, H represents pork ham, J represents pork shoulder, and W represents pork belly).

**Figure 4 molecules-28-01726-f004:**
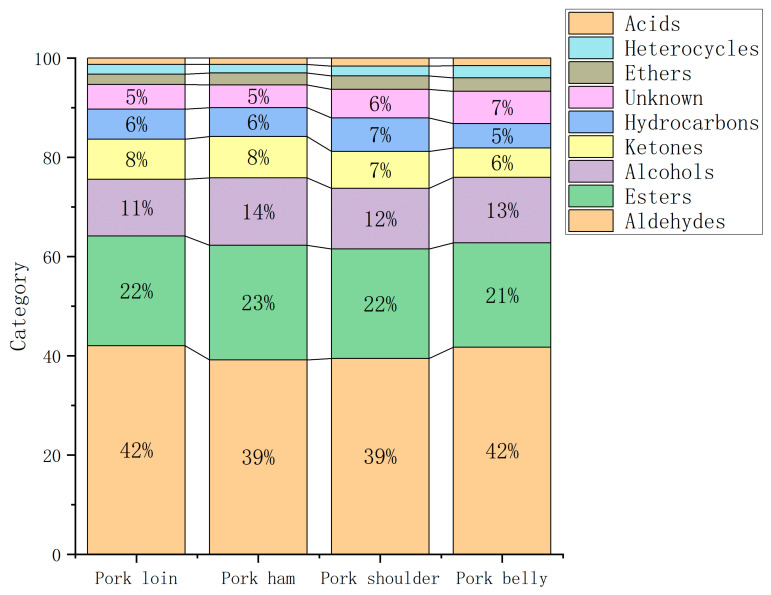
Proportions of volatile organic compounds in four muscles of pork.

**Figure 5 molecules-28-01726-f005:**
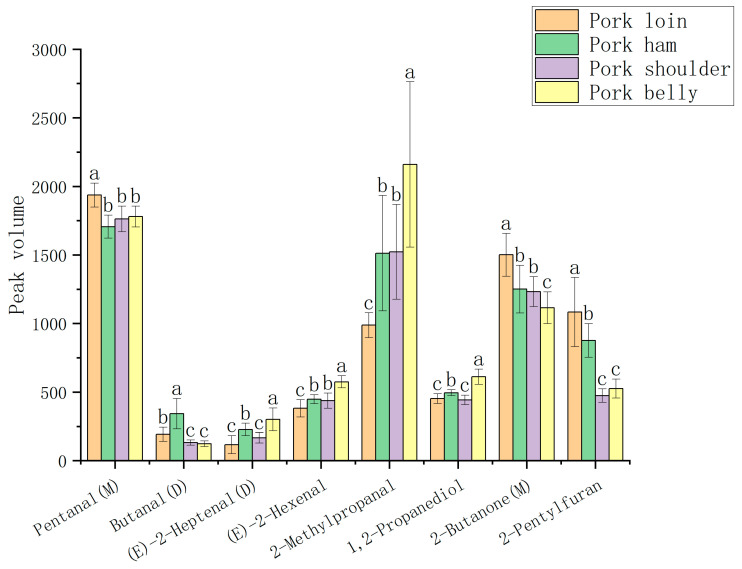
Significant differences of certain compounds in four muscles of pork (note: different letters (a, b, c) indicate a significant difference (*p* < 0.05) according to one-way analysis of variance).

**Figure 6 molecules-28-01726-f006:**
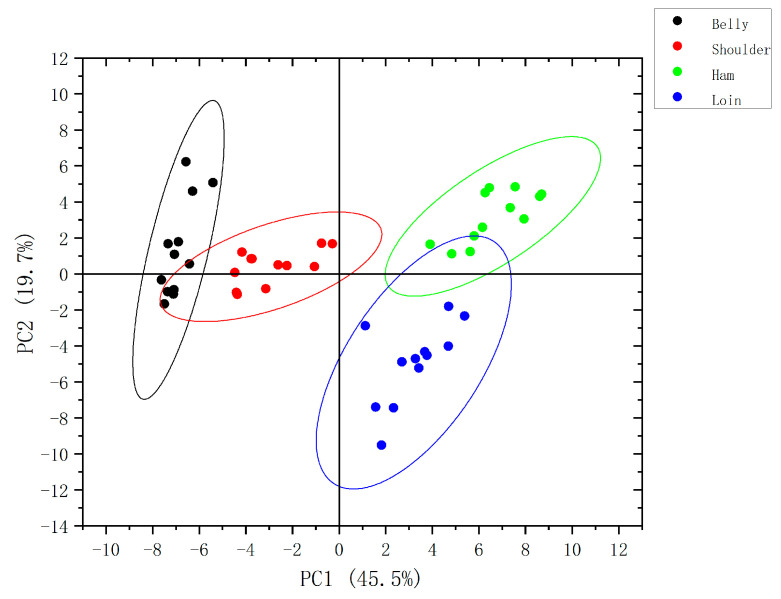
Plot of the principal component analysis scores of volatile organic compounds in four muscles of pork.

**Figure 7 molecules-28-01726-f007:**
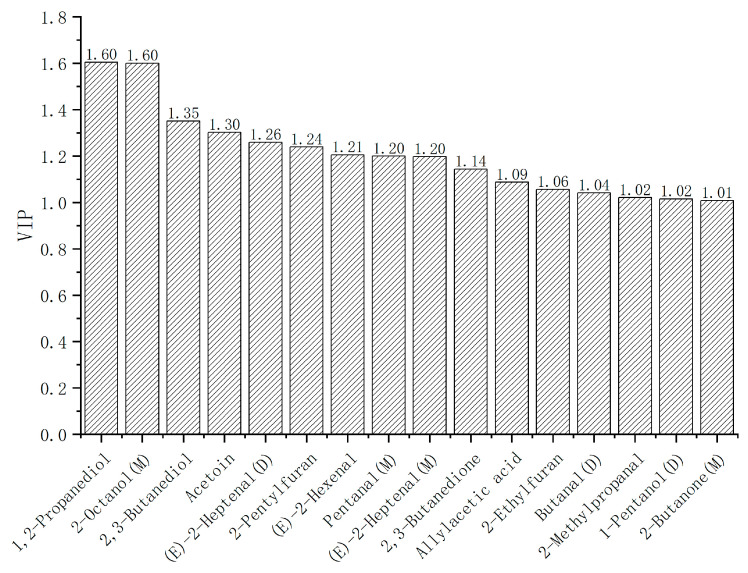
Volatile organic compounds with VIP > 1.

**Table 1 molecules-28-01726-t001:** Qualitative analysis of pork volatile organic compounds.

Category	Compound	CAS#	Formula	MW	RI	Rt [sec]	Dt [a.u.]	Compound Peak Volume			
								Pork Loin	Pork Ham	Pork Shoulder	Pork Belly
Aldehydes	Benzaldehyde (M)	C100527	C7H6O	106.1	974.6	542.409	1.15333	2120.21 ± 296.54 ^a^	1863.12 ± 208.26 ^b^	797.13 ± 305.38 ^c^	378.67 ± 74.79 ^d^
	Benzaldehyde (D)	C100527	C7H6O	106.1	973.4	540.193	1.4786	578.11 ± 162.20 ^a^	449.27 ± 117.31 ^b^	183.14 ± 50.77 ^c^	138.35 ± 7.76 ^c^
	Pentanal (M)	C110623	C5H10O	86.1	711.2	196.868	1.18661	1936.28 ± 87.90 ^a^	1706.06 ± 83.63 ^b^	1762.36 ± 93.79 ^b^	1779.76 ± 75.81 ^b^
	Pentanal (D)	C110623	C5H10O	86.1	703.7	190.353	1.41154	9512.89 ± 810.42 ^b^	10,457.37 ± 540.91 ^a^	8303.84 ± 617.33 ^c^	7102.84 ± 286.04 ^d^
	Heptanal	C111717	C7H14O	114.2	899.2	404.089	1.32831	3335.29 ± 216.03 ^a^	3215.26 ± 82.00 ^a^	2433.27 ± 186.40 ^b^	2158.79 ± 83.56 ^c^
	Butanal (M)	C123728	C4H8O	72.1	606.2	137.56	1.11902	474.47 ± 159.96 ^b^	591.45 ± 106.56 ^a^	524.42 ± 89.19 ^ab^	442.03 ± 110.91 ^b^
	Butanal (D)	C123728	C4H8O	72.1	615.8	142.193	1.2939	192.78 ± 52.02 ^b^	343.19 ± 110.69 ^a^	131.99 ± 18.43 ^c^	123.53 ± 21.11 ^c^
	Octanal	C124130	C8H16O	128.2	1012.8	614.425	1.40382	3383.74 ± 525.26 ^a^	3529.11 ± 317.77 ^a^	1535.75 ± 267.42 ^b^	1289.57 ± 113.11 ^b^
	Nonanal (M)	C124196	C9H18O	142.2	1107.1	799.01	1.47879	3940.39 ± 847.84 ^b^	4871.14 ± 901.05 ^a^	1198.93 ± 221.97 ^c^	982.29 ± 87.26 ^c^
	Nonanal (D)	C124196	C9H18O	142.2	1105.3	795.367	1.95105	742.92 ± 331.94 ^b^	1183.82 ± 439.98 ^a^	183.45 ± 14.91 ^c^	180.67 ± 15.32 ^c^
	(E)-2- Heptenal(M)	C18829555	C7H12O	112.2	960.7	516.927	1.26362	1320.8 ± 371.88 ^b^	1922.3 ± 186.83 ^a^	1515.6 ± 196.52 ^b^	1897.11 ± 225.66 ^a^
	(E)-2- Heptenal (D)	C18829555	C7H12O	112.2	956.7	509.605	1.67358	116.75 ± 65.43 ^c^	227.22 ± 45.86 ^b^	165.81 ± 37.73 ^c^	301.54 ± 83.18 ^a^
	(*E*, *E*)-2,4-Heptadienal	C4313035	C7H10O	110.2	961.7	518.81	1.62407	100.73 ± 33.88 ^a^	99.90 ± 25.61 ^a^	60.83 ± 12.96 ^b^	37.30 ± 4.19 ^c^
	3-Methyl- butanal (M)	C590863	C5H10O	86.1	667.2	166.95	1.1779	2887.51 ± 286.52 ^a^	2823.69 ± 210.76 ^a^	2222.76 ± 146.65 ^b^	1819.61 ± 96.37 ^c^
	3-Methyl- butanal (D)	C590863	C5H10O	86.1	666.1	166.44	1.40598	2181.52 ± 630.74 ^a^	1990.03 ± 596.70 ^a^	808.87 ± 171.68 ^b^	436.63 ± 70.17 ^c^
	Hexanal	C66251	C6H12O	100.2	828.6	314.172	1.25408	7895.20 ± 724.88 ^b^	9235.53 ± 491.80 ^a^	6985.59 ± 350.31 ^c^	6038.28 ± 287.44 ^d^
	(*E*)-2- Hexenal	C6728263	C6H10O	98.1	847.8	337.473	1.18775	381.43 ± 62.38 ^c^	447.91 ± 32.08 ^b^	437.32 ± 55.23 ^b^	574.74 ± 44.25 ^a^
	2-Methylpropanal	C78842	C4H8O	72.1	555.9	113.276	1.09944	987.87 ± 90.39 ^c^	1512.66 ± 420.43 ^b^	1521.93 ± 345.19 ^b^	2160.05 ± 602.78 ^a^
Alcohols	1- Hexanol (M)	C111273	C6H14O	102.2	874.7	370.068	1.32792	354.74 ± 120.53 ^b^	427.43 ± 49.74 ^a^	260.48 ± 24.04 ^c^	221.97 ± 19.44 ^c^
	1-Hexanol (D)	C111273	C6H14O	102.2	888.4	386.702	1.63021	257.45 ± 89.91 ^b^	622.01 ± 189.54 ^a^	100.69 ± 27.04 ^c^	67.12 ± 13.39 ^c^
	1-Octen-3-ol	C3391864	C8H16O	128.2	1001.2	591.772	1.16155	3323.17 ± 610.15 ^b^	5756.30 ± 910.10 ^a^	2329.79 ± 365.16 ^c^	1726.14 ± 184.19 ^d^
	2-Octanol (M)	C123966	C8H18O	130.2	990.3	571.216	1.45616	583.41 ± 60.03 ^b^	926.83 ± 73.73 ^a^	905.51 ± 88.45 ^a^	530.99 ± 82.30 ^b^
	2-Octanol (D)	C123966	C8H18O	130.2	1011.1	611.102	1.83377	1896.67 ± 696.96 ^b^	2405.66 ± 503.10 ^a^	479.15 ± 116.43 ^c^	344.29 ± 43.00 ^c^
	2,3- Butanediol	C513859	C4H10O2	90.1	805.4	286.059	1.35755	877.61 ± 63.95 ^c^	1135.21 ± 49.10 ^a^	1104.73 ± 68.45 ^a^	950.92 ± 106.48 ^b^
	1,2- Propanediol	C57556	C3H8O2	76.1	742.2	223.859	1.12866	452.53 ± 35.50 ^c^	495.06 ± 21.68 ^b^	442.61 ± 35.21 ^c^	611.89 ± 55.71 ^a^
	Isopropyl alcohol	C67630	C3H8O	60.1	508.1	90.245	1.08847	7.54 ± 1.04 ^b^	13.59 ± 3.06 ^ab^	24.21 ± 16.41 ^a^	26.18 ± 24.86 ^a^
	1- Pentanol (M)	C71410	C5H12O	88.1	781.1	257.677	1.25906	2399.79 ± 212.89 ^b^	2492.52 ± 145.90 ^ab^	2379.84 ± 205.06 ^b^	2573.71 ± 96.58 ^a^
	1- Pentanol (D)	C71410	C5H12O	88.1	779.6	256.306	1.50174	1295.36 ± 343.15 ^b^	1857.61 ± 387.71 ^a^	1491.68 ± 271.33 ^b^	1772.76 ± 173.97 ^a^
Ketones	3-Octanone	C106683	C8H16O	128.2	991.3	573.062	1.72909	1117.15 ± 346.98 ^b^	2344.04 ± 572.68 ^a^	649.53 ± 175.88 ^c^	412.05 ± 29.15 ^c^
	2-Heptanone	C110430	C7H14O	114.2	890.3	388.894	1.25915	834.42 ± 140.12 ^b^	1187.69 ± 161.84 ^a^	574.76 ± 94.44 ^c^	425.44 ± 38.64 ^d^
	2,3- Butanedione	C431038	C4H6O2	86.1	605.4	137.132	1.16982	559.32 ± 109.36 ^b^	714.50 ± 59.97 ^a^	725.68 ± 68.38 ^a^	600.35 ± 94.40 ^b^
	Acetoin	C513860	C4H8O2	88.1	737.6	219.882	1.33962	94.23 ± 19.88 ^b^	334.86 ± 108.05 ^a^	317.67 ± 162.67 ^a^	128.34 ± 18.96 ^b^
	2- Butanone (M)	C78933	C4H8O	72.1	611.2	139.959	1.06941	1500.92 ± 157.12 ^a^	1250.83 ± 173.75 ^b^	1232.74 ± 108.98 ^b^	1114.81 ± 116.07 ^c^
	2- Butanone (D)	C78933	C4H8O	72.1	607.7	138.252	1.25195	3979.22 ± 1258.45 ^a^	4055.29 ± 666.62 ^a^	2304.2 ± 834.38 ^b^	1264.30 ± 506.70 ^b^
Esters	Ethyl butyrate	C105544	C6H12O2	116.2	800	279.461	1.56537	15,609.94 ± 1092.87 ^b^	17,857.87 ± 984.90 ^a^	13,608.52 ± 679.67 ^c^	11,597.96 ± 560.99 ^d^
	Butanoic acid 3-methylethyl ester	C108645	C7H14O2	130.2	929.3	459.215	1.25444	1646.54 ± 266.32 ^b^	2279.87 ± 310.72 ^a^	1239.95 ± 167.13 ^c^	912.01 ± 78.93 ^d^
	Ethyl hexanoate	C123660	C8H16O2	144.2	989.5	569.738	1.80012	491.63 ± 142.28 ^b^	1286.99 ± 334.40 ^a^	384.96 ± 160.01 ^b^	118.58 ± 22.91 ^c^
	Ethyl acetate	C141786	C4H8O2	88.1	625.7	146.922	1.34904	64.62 ± 11.87 ^b^	81.79 ± 16.53 ^a^	69.04 ± 3.98 ^b^	67.24 ± 7.25 ^b^
	Ethyl pentanoate	C539822	C7H14O2	130.2	900.5	406.4	1.70779	4345.36 ± 1108.18 ^b^	5858.13 ± 746.71 ^a^	1919.29 ± 467.12 ^c^	1354.86 ± 185.53 ^c^
Ethers	1,2- Dimethoxyethane	C110714	C4H10O2	90.1	629.5	148.785	1.1013	77.78 ± 20.41 ^b^	116.93 ± 21.20 ^a^	67.30 ± 19.76 ^bc^	56.18 ± 19.96 ^c^
	Tert-butyl methyl ether	C1634044	C5H12O	88.1	545.2	108.136	1.13786	2010.77 ± 463.89 ^b^	2764.82 ± 532.35 ^a^	2037.07 ± 612.05 ^b^	1761.61 ± 510.25 ^b^
Acids	Allylacetic acid	C591800	C5H8O2	100.1	900.7	406.838	1.42289	128.02 ± 7.83 ^c^	162.15 ± 19.97 ^b^	186.51 ± 19.25 ^a^	181.14 ± 19.23 ^a^
	Propanoic acid	C79094	C3H6O2	74.1	668.4	167.518	1.27935	417.70 ± 68.18 ^b^	542.43 ± 66.51 ^a^	380.22 ± 69.31 ^b^	317.52 ± 62.70 ^c^
	2-Methylpropanoic acid	C79312	C4H8O2	88.1	748.5	229.282	1.17107	729.53 ± 55.21 ^b^	853.77 ± 33.10 ^a^	700.38 ± 99.11 ^b^	530.17 ± 32.81 ^c^
Heterocycles	2-Ethylfuran	C3208160	C6H8O	96.1	707.6	193.807	1.29263	909.65 ± 64.54 ^b^	1122.93 ± 61.20 ^a^	1087.95 ± 61.96 ^a^	1114.27 ± 111.51 ^a^
	2-Pentylfuran	C3777693	C9H14O	138.2	996.1	581.926	1.26175	1083.64 ± 251.73 ^a^	876.95 ± 122.49 ^b^	474.46 ± 49.55 ^c^	525.73 ± 69.32 ^c^
Hydrocarbons	2,2,4,6,6- Pentamethyl heptane	C13475826	C12H26	170.3	991.3	573.062	1.37018	3442.20 ± 482.06 ^b^	4530.26 ± 220.91 ^a^	2870.55 ± 534.99 ^c^	1293.13 ± 249.67 ^d^
	Beta- ocimene	C13877913	C10H16	136.2	1041.9	671.3	1.25802	493.07 ± 109.19 ^b^	735.96 ± 71.70 ^a^	359.82 ± 67.10 ^c^	300.38 ± 32.46 ^c^
	Alpha- phellandrene	C99832	C10H16	136.2	1007.5	604.085	1.67301	2140.81 ± 606.39 ^a^	1641.64 ± 423.30 ^b^	2053.57 ± 262.20 ^a^	1679.67 ± 174.57 ^b^

Note: Values in the same row are marked with different letters, indicating significant differences among the four parts (*p* < 0.05).

## Data Availability

The data presented in this study are available on request from the corresponding author. The data are not publicly available due to involving other research content.

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
