# Peer review of "Analysis of the Differences in Volatile Organic Compounds in Different Muscles of Pork by GC-IMS"

_molecules, 2023, doi:10.3390/molecules28041726_

Round 1
Reviewer 1 Report
The volatile flavor differences of the four pork muscles were analyzed and characteristic flavor substances were determined in this study. It was a valuable research. To polish this paper, some details should be modified or supplemented. Minor revisoin should be done.
In Figure 4. The colors for “Acids” and “Aldhydes” are the same.
Line 92. Is there NIST database for GG-IMS? Please check this
Line 127-129. What is the basis for the statement? Is there any reference?
Line 129. “benzaldehyde has an unpleasant taste” What is the taste of benzaldehyde? Is there nonvolatile flavor of benzaldehyde?
Line 142-143. Is there any reference for this statement?
Line 157. Why did the authors state that “they (heterocyclic compounds) can promote the formation of pork flavor and make pork flavor diversified”
Line 212-215. Why the samples were heated in the SPME vial? The air in the vial is limited. The reaction in the vial would be different from reactions in air or in other containers.
Please check the spellings of “2-pentyl furan”, “2-Ethyl furan”, “Allyl acetic acid” and “2-methyl propanal” in Figure 5, Figure 7 and in the main body.
More results and discussion should be supplemented. The analysis of VIP and the contents of compounds in Line 188-199 was insufficient to verify the characteristic flavor compounds.
What is the purpose to do the PCA analysis? What is the relationship between the PCA analysis and verification of the characteristic flavor compounds?
How the 16 compounds were selected on the basis of Figure 6 and 7?
Please check the grammar of this paper, especially the tense.
Some statements should refer to references. Please check the reference citing.
Reviewer 2 Report
1. In Figure 3, what do B, H, J, W mean?
2. Methods are presented in sufficient detail for replication; however, the use of standards in the HS-GC-MS analysis needs to be clarified. Which standards were used? Was an internal standard added to ensure no losses of bioactive compounds in the extraction process?
3. More information about PCA and PLS-DA are lacking. Please add the matrix data set, the data pre-treatment and a brief theory.
4. Table S1 should be described in the main text.
